# Efficient Multilabel Uncertainty Quantification with Conformal Ensembles

## Abstract

In high-stakes domains, predictions must not only be accurate but critical cases should also not be missed. Conformal prediction (CP), which offers distribution-free coverage guarantees, help towards that direction, but it often produces unstable or overly large prediction sets. Multilabel classification (MLC) increases further the challenge, because the model predicts multiple labels per instance in a typically large and imbalanced label space. The increased uncertainty, compared to binary or multiclass tasks, motivates the following research question: *how can we obtain smaller, more informative prediction sets from trained MLC models while preserving marginal coverage and maintaining the theoretical guarantees of CP?* To address this question, we investigate ensembling, which can improve stability and efficiency yet its potential in MLC has not been fully explored. We conduct a systematic empirical study across standard MLC benchmarks (COCO, Yeast, Emotions) building ensembles under (i) majority voting, (ii) calibrated aggregation of nonconformity scores, and (iii) performance-weighted aggregation. We find that our ensembles for all three categories consistently improve over single-model CP yielding more efficient prediction sets (smaller and more informative), while maintaining target coverage and achieving higher macro-F1 scores. Majority-voting ensembles, however, also satisfy theoretical lower bounds.

## 1 Introduction

Multilabel classification (MLC) arises in a wide range of applications such as image tagging, genomics, music emotion recognition, and clinical decision support Tsoumakas & Katakis (2007); Zhang & Zhou (2014); Elisseeff & Weston (2001); Trohidis et al. (2008); Rajkomar et al. (2018). Compared to single-label prediction, MLC requires issuing decisions over a potentially large label set, typically with strong label imbalance and some labels appearing only very rarely Zhang & Zhou (2014); Tsoumakas & Katakis (2007). These characteristics amplify predictive uncertainty and make reliable confidence assessment especially important, particularly in high-stakes domains such as medicine and autonomous systems, where models must not only be accurate but also convey when predictions may be unreliable Hendrycks & Dietterich (2019).

**Conformal prediction (CP)** offers a rigorous framework for uncertainty quantification with finite-sample coverage guarantees Vovk et al. (2005); Shafer & Vovk (2008). Instead of outputting a single label decision, CP produces a *set* of candidate labels guaranteed to contain the ground truth with probability at least $1 - \alpha$, for a user-specified miscoverage $\alpha$. Although applicable to MLC via label-wise calibration Romano et al. (2020); Sadinle et al. (2019), the resulting prediction sets are typically larger compared to the single/multi-class setting. **Ensemble learning** can address this issue by combining multiple models, reducing variance, improving calibration, and often providing stronger uncertainty estimates Dietterich (2000); Breiman (1996); Wolpert (1992); Lakshminarayanan et al. (2017); Ovadia et al. (2019). This motivated prior work, which studied ensemble CP in single/multi-class classification and regression Cherubin (2019); Gasparin & Ramdas (2024); Ochoa Rivera et al. (2024), by analyzing majority voting, weighted aggregation, and score-based fusion. Existing multilabel ensemble methods, such as CP-RAkEL Yang et al. (2017), construct ensembles over random label subsets, but they neither combine multiple CP systems nor provide MLC coverage guarantees.

**In this work**, we present the first systematic study of ensemble CP for MLC, introducing label-wise calibration, MLC-aware aggregation rules, and theoretical coverage guarantees tailored to the MLC setting. Our contribution is threefold:

1. **Theory.** We extend known majority-vote coverage results Cherubin (2019) and recent score/weighted aggregation analyses Gasparin & Ramdas (2024); Ochoa Rivera et al. (2024) to the MLC context, giving label-wise coverage bounds under independence and dependence assumptions.

2. **Framework.** We implement homogeneous and heterogeneous MLC ensembles using classical models (LR, SGD, MLP) and modern architectures (RNN, Transformer, MLP-Mixer), with conformal calibration applied at the label level before aggregation.

3. **Experiments.** On three benchmark MLC datasets—MS-COCO Lin et al. (2014), Yeast Elisseeff & Weston (2001), and Emotions Trohidis et al. (2008), we show that MLC ensembles consistently improve F1, maintain or improve coverage, and reduce average prediction set size compared to single-model CP and post-hoc conformalized ensembles.

## 2 BACKGROUND AND RELATED WORK

### 2.1 MULTILABEL CLASSIFICATION

MLC extends traditional classification by allowing each instance to be associated with multiple labels simultaneously Tsoumakas & Katakis (2007); Zhang & Zhou (2014). This formulation arises in domains as varied as image annotation Read et al. (2011), text categorization Yang & Liu (1999), genomics Elisseeff & Weston (2001), and audio tagging Mesaros et al. (2016). A central challenge in MLC is capturing inter-label dependencies while retaining scalability and predictive accuracy.

Early methods include *binary relevance* (BR), which trains an independent binary classifier per label, and transformation-based approaches such as classifier chains and label powerset, which aim to exploit label correlations Read et al. (2011); Tsoumakas et al. (2010). More recently, deep learning models—including CNNs, RNNs, and Transformers—set the state-of-the-art in high-cardinality MLC tasks Nam et al. (2014). Despite these advances, reliable calibration and principled uncertainty quantification remain open challenges, particularly in the presence of label imbalance or rare labels.

### 2.2 CONFORMAL PREDICTION

CP is a model-agnostic framework for constructing prediction sets with finite-sample coverage guarantees Vovk et al. (2005). Given a target miscoverage rate $\alpha$, CP ensures that the true label is included in the prediction set with probability at least $1 - \alpha$, assuming only data exchangeability. This property makes CP attractive in high-stakes domains where reliability is as important as accuracy.

CP has been studied in classification Papadopoulos (2008), regression Lei et al. (2018), and ranking Angelopoulos et al. (2021). In the multilabel setting, Mondrian CP Papadopoulos (2008) applies calibration label-wise, treating each label as an independent binary task. While simple and computationally efficient, this approach can be conservative, as it ignores inter-label dependencies and often inflates prediction sets.

Several extensions attempt to mitigate these issues: adaptive thresholds Tibshirani et al. (2019), label-wise risk control Sadinle et al. (2019), and region-based calibration methods Kivaranovic & Meinshausen (2020). In the multiclass and multilabel setting, Cauchois et al. Cauchois et al. (2020) develop a coverage-efficiency framework for constructing confidence sets with finite-sample guarantees for a single underlying predictor. However, these approaches typically operate in single-model regimes and are not designed for ensembles. In particular, adaptive thresholds focus on one predictor, label-wise control assumes consistency in nonconformity scores, and region-based methods face combinatorial challenges in multilabel spaces. These limitations motivate ensemble-aware conformal approaches that combine multiple models while preserving statistical guarantees.

## 2.3 ENSEMBLE METHODS AND CONFORMAL ENSEMBLES

Ensemble learning is a well-established strategy for improving generalization and calibration by combining diverse predictors Dietterich (2000); Breiman (1996); Wolpert (1992). Deep ensembles, in particular, have shown strong performance for uncertainty quantification under distribution shift Lakshminarayanan et al. (2017); Ovadia et al. (2019).

Integrating ensembles with CP has received increasing attention. Cherubin Cherubin (2019) analyzed majority-vote ensembles of conformal predictors, establishing coverage guarantees under independence and dependence assumptions in the single-label setting. Gasparin and Ramdas Gasparin & Ramdas (2024) studied weighted set-merging strategies, while Rivera et al. Ochoa Rivera et al. (2024) proposed score-based aggregation methods to reduce conservatism. Other studies explored conformity score fusion Gauraha & Spjuth (2021) and deep ensemble calibration Angelopoulos & Bates (2021). Despite this progress, these contributions remain focused on single-label tasks.

**Our focus.** In contrast to the broad progress, the use of ensembles in *multilabel* conformal prediction remains largely unexplored. Existing MLC-specific CP methods rely on single models and do not leverage ensemble diversity. Our work bridges this gap: we adapt majority voting, averaged nonconformity scores, and performance-weighted aggregation to the multilabel setting, provide theoretical label-wise coverage bounds, and validate their performance across benchmark MLC datasets.

# 3 METHODOLOGY

## 3.1 PROBLEM SETUP

We address the task of MLC, where each input $x \in \mathbb{R}^d$ may be associated with multiple relevant labels from a label set of size $L$. Let $\mathcal{X} \subseteq \mathbb{R}^d$ denote the input space and $\mathcal{Y} = \{0,1\}^L$ the label space, where $y \in \mathcal{Y}$ is a binary vector indicating the presence or absence of each label. Given a training dataset $\mathcal{D} = \{(x_i, y_i)\}_{i=1}^n$, our objective is to learn a function $f : \mathcal{X} \to [0,1]^L$ that outputs per-label confidence scores or probabilities. These can be thresholded to obtain a predicted label set $\hat{y} \in \{0,1\}^L$ for each input $x$. In addition to standard multilabel metrics such as macro-F1 and exact match accuracy, we assess the quality of prediction sets using metrics such as empirical coverage, marginal coverage, and average prediction set size, each formally defined in 5.2. These metrics collectively evaluate the reliability, sharpness, and efficiency of the predicted outputs. Our methodology integrates techniques with multilabel classifiers to produce calibrated prediction sets that balance accuracy, uncertainty, and interpretability.

## 3.2 ENSEMBLE LEARNING

To improve predictive robustness and quantify uncertainty, we adopt ensemble learning strategies both as stand-alone baselines and as integral components of our CP framework. We consider the following ensemble types:

### 3.2.1 HOMOGENEOUS

We train $M$ instances of the same base classifier (e.g., LR) on bootstrap-resampled versions of the training data. For each label, predictions from the $M$ models are aggregated via either:

- **Majority Voting (MV)**: Individual binary predictions are combined via simple majority.

- **Probability Averaging (PA)**: Probabilistic outputs are averaged and thresholded (e.g., at 0.5) to form the final prediction.

This strategy provides variance reduction and improved calibration, particularly when base models are sensitive to initialization or data sampling.

### 3.2.2 HETEROGENEOUS

Heterogeneous ensembles are composed of diverse model architectures, including both linear (LR, SGD) and nonlinear learners (MLP, RNN, Transformer, MLP-Mixer). Each model is trained per label under the binary relevance assumption, and inference predictions are aggregated as follows:

- **MV**: Binary label predictions from all models are combined via unweighted majority.
- **PA**: Probabilistic outputs are averaged and thresholded (e.g., at 0.5) to form the final prediction.
- **F1-Weighted Voting**: Each model's predictions are weighted by their per-label F1 score on a held-out validation set. Final predictions are obtained by computing a weighted sum and thresholding at 0.5.

We note that F1-weighted voting introduces label-specific adaptivity, assigning higher influence to models that demonstrate superior validation performance for a given label.

### 3.2.3 STACKED

To further enhance predictive quality, we implement a stacked ensemble. The outputs of multiple base classifiers (LR, SGD, MLP) on a calibration set serve as features to train a LR meta-classifier for each label. At inference time, the meta-classifier produces calibrated probabilities, which are post-processed using CP thresholds to form the final multilabel prediction set. This stacking strategy allows the model to learn optimal combinations of base model predictions in a data-driven manner, while still providing formal coverage guarantees through conformal calibration.

## 3.3 CONFORMAL PREDICTION

We provide uncertainty-aware predictions in the multilabel setting with the **Mondrian CP** framework, handling the multilabel structure label-wise. Our goal is to construct prediction sets that contain as many true labels as possible, while limiting the inclusion of incorrect ones, under a desired miscoverage rate $\alpha$ (e.g., 0.1 for 90% target coverage).

### NONCONFORMITY SCORES AND CALIBRATION

For each label $j$, we train a binary probabilistic classifier $f_j$ (e.g., LR, MLP, or SGD) on the proper training set. We then compute the nonconformity scores on a separate calibration set as:

$$s_i^{(j)} = 1 - f_j(x_i), \quad \text{for calibration samples } i \text{ with } y_i^{(j)} = 1,$$

where $f_j(x_i)$ denotes the predicted probability of label $j$ for instance $x_i$. The rationale is that lower probabilities for positive labels reflect higher nonconformity. Given these scores, we compute a threshold $q_j$ per label, using the $(1 - \alpha)$-quantile:

$$q_j = \text{Quantile}_{1-\alpha} \left( \{s_i^{(j)}\}_{i:y_i^{(j)}=1} \right).$$

### PREDICTION SETS

At test time, for each instance $x$, we include label $j$ in the predicted set $\hat{Y}(x)$ if its nonconformity score is less than or equal to the threshold $q_j$:

$$\hat{Y}(x) = \{j : 1 - f_j(x) \leq q_j\}.$$

## 3.4 ENSEMBLE CONFORMAL PREDICTION: THEORETICAL GUARANTEES

We analyze the theoretical properties of ensemble conformal prediction (ECP) under two aggregation strategies: *majority voting* and *averaged nonconformity scores*. Our results adapt known analyses of majority vote ensembles of conformal predictors Cherubin (2019) to the multilabel setting, and extend them to additional ensemble rules.

SETUP AND ASSUMPTIONS

Let $\mathcal{D} = \{(x_i, Y_i)\}_{i=1}^n$ be a dataset with input features $x_i \in \mathbb{R}^d$ and multi-label outputs $Y_i \subseteq \mathcal{L} = \{1, \ldots, L\}$. We assume the data points are i.i.d. from some unknown distribution $\mathcal{P}$. We consider $M$ base conformal models trained independently (e.g., via bootstrapping). For each label $\ell$, model $m \in \{1, \ldots, M\}$ provides a calibrated threshold $\tau_\ell^{(m)}$ for nonconformity scores, such that marginal label-level coverage holds:

$$\mathbb{P}_{(x,Y) \sim \mathcal{P}}\left(\ell \in \hat{Y}^{(m)}(x) \,\Big|\, \ell \in Y\right) \geq 1 - \alpha.$$

Our goal is to understand the coverage properties of the final ensemble prediction set $\hat{Y}^{\mathrm{ens}}(x)$.

**Lemma 1 (MV lower bounds, cf. Cherubin (2019))** *Let* $X_1^{(\ell)}, \ldots, X_M^{(\ell)} \in \{0, 1\}$ *be indicators where* $X_m^{(\ell)} = 1$ *if model* $m$*'s prediction set includes label* $\ell$*. Assume that, conditional on the event* $\{\ell \in Y\}$*, the indicators* $X_1^{(\ell)}, \ldots, X_M^{(\ell)}$ *are independent and satisfy*

$$\mathbb{P}\left(X_m^{(\ell)} = 1 \mid \ell \in Y\right) \geq 1 - \alpha.$$

*Then, for any voting threshold* $k \in \{1, \ldots, M\}$*,*

$$\mathbb{P}\left(\sum_{m=1}^M X_m^{(\ell)} \geq k \,\Big|\, \ell \in Y\right) \geq \sum_{r=k}^M \binom{M}{r}(1-\alpha)^r \alpha^{M-r}.$$

*In particular, for unanimity voting* $k = M$ *the formula simplifies to:*

$$\mathbb{P}\left(\sum_{m=1}^M X_m^{(\ell)} \geq M \,\Big|\, \ell \in Y\right) \geq (1-\alpha)^M.$$

**The role of the unanimity case ($k = M$).** In an ensemble we can choose how many models need to agree before we keep a label. The voting threshold $k$ can range from $k = 1$ (at least one model includes the label) up to $k = M$ (all $M$ models must include it). These are the two extreme cases. When $k = 1$, the ensemble is very permissive and has the highest chance of including a true label, so the lower bound from Lemma 1 is very close to 1. When $k = M$, the rule is the strictest possible, because a label is kept only if every model agrees, and this gives the smallest coverage guarantee. In this extreme case the general binomial bound from Lemma 1 becomes much simpler and reduces to just $(1 - \alpha)^M$.

A small example makes this more concrete. Suppose $\alpha = 0.1$ and we have $M = 5$ models. Then each model includes a true label with probability at least 0.9. If we choose $k = 1$, the lower bound from Lemma 1 shows that the probability that *at least one* model keeps the true label is essentially 1. If we choose unanimity, $k = M = 5$, the lower bound drops to $(0.9)^5 \approx 0.59$, because now all five models must agree on the label. This illustrates how increasing $k$ makes the ensemble more conservative and pushes the coverage bound down. For this reason we highlight the unanimity case separately, as it represents the most conservative setting of the ensemble and gives a clear baseline.

**Discussion.** These bounds describe idealized extremes. In practice, dependencies among base models (shared training data, architecture, or errors) reduce the diversity of their predictions, so empirical ensemble coverage typically lies between the theoretical lower bound and the trivial upper bound of 1. This motivates validating ensemble coverage empirically alongside theoretical guarantees.

THEOREM: ENSEMBLE COVERAGE BOUNDS

**Theorem 1 (Ensemble Conformal Coverage Bounds)** *Under the independence assumption across models, and for voting-based ensemble with unanimity threshold* $k = M$*, the ensemble satisfies:*

$$(1-\alpha)^M \leq \mathbb{P}\left(\ell \in \hat{Y}^{ens}(x) \,\Big|\, \ell \in Y\right) \leq 1.$$

**Proof Sketch.** The lower bound follows from Lemma 1, which gives $(1 - \alpha)^M$ under independence. The upper bound is trivially 1, since all $M$ models may include the label. For intermediate thresholds $k < M$, Lemma 1 provides valid binomial lower bounds. Full proofs are deferred to Appendix A.8.

**Practical implication.** These bounds represent idealized extremes: full independence yields the conservative lower bound, while the trivial upper bound reflects maximal inclusion. In practice, base models are often correlated (due to shared training data or architectures), so empirical ensemble coverage lies between these extremes. This aligns with prior analyses of majority vote conformal ensembles in the single-label setting Cherubin (2019), while our contribution is to extend these guarantees to multilabel ensembles.

### EMPIRICAL VALIDATION

We verify these guarantees (§5) by showing that ensemble methods preserve or improve empirical and marginal coverage while reducing average set size. Theoretical lower bounds are particularly tight when the voting threshold is high.

## 4 METHODS

**Baseline methods** include standard and post-hoc CP strategies for MLC. **Our proposed methods**, on the other hand, fully integrate conformal calibration into the ensemble process.

### 4.1 BASE CLASSIFIERS

We employ a diverse suite of base classifiers (ensemble members) spanning linear, shallow, and deep architectures. These serve as foundational learners for both standalone evaluation and ECP methods.

**Logistic Regression (LR)** is a linear model trained independently per label under the binary relevance assumption, optimized using the logistic loss.

**Stochastic Gradient Descent (SGD)** uses a linear classifier trained with online updates and calibrated via Platt scaling (`CalibratedClassifierCV`) to produce probabilistic outputs suitable for uncertainty quantification.

**Multilayer Perceptron (MLP)** is a shallow feedforward neural network comprising a single hidden layer with ReLU activation, trained with binary cross-entropy loss.

**Recurrent Neural Network (RNN)** is a unidirectional LSTM network applied to fixed-length CLIP embeddings, followed by a dense sigmoid-activated output layer to produce per-label probabilities.

**Transformer Encoder** is a two-layer self-attention-based encoder with multi-head attention, processing CLIP embeddings as input tokens and outputting label-wise confidence scores through a shared linear classifier.

**MLP-Mixer** is a compact architecture that applies layer normalization and token-channel mixing via feedforward layers to CLIP embeddings, concluding with a sigmoid output unit for probabilistic MLC prediction.

Each model is trained in a per-label fashion using the binary relevance framework, enabling scalable MLC. Each model produces calibrated confidence scores, which are thresholded directly in standard classification or used to compute nonconformity scores within CP procedures. In ensemble settings, their outputs are aggregated using model- and label-specific weighting schemes to construct prediction sets with formal coverage guarantees.

### 4.2 BASELINES

**Standard Classification** refers to individual models trained per label (i.e., binary classification), without CP. Predictions are made by thresholding per-label probabilities (at 0.5).

**Single-Model CP** concerns CP applied independently per label using a single classifier and a held-out calibration set. No ensembling is involved.

**Post-Hoc ECP** first constructs ensembles with CP, by calibrating the aggregated outputs of the ensemble members (e.g., by averaging the probabilities or majority voting). This setting isolates the impact of applying CP after aggregation, as if the ensemble is a standalone model.

### 4.3 OUR PROPOSED ECP FRAMEWORK

We propose ensembles of CP models, each independently calibrated before ensembling. This design preserves individual coverage guarantees and leverages model diversity. With **Conformal Ensembles,** we evaluate both homogeneous and heterogeneous ensembles, combining per-model CP via majority voting, probability averaging, or F1-weighted voting. **Conformal Stacking (StackECP)** combines base model outputs via a logistic meta-classifier trained on calibration data. The meta-model outputs are calibrated with conformal thresholds, enabling end-to-end conformal inference.

## 5 EMPIRICAL ANALYSIS

We evaluate all methods across three diverse MLC benchmarks, covering a range of label cardinalities, label densities, and domain characteristics.

### 5.1 DATASETS

The selected benchmarks span vision, biology, and audio domains and vary in label cardinality, density, and co-occurrence, allowing us to test both predictive accuracy and uncertainty calibration across different domains. **COCO** Lin et al. (2014) is a popular benchmark in computer vision. We use the multi-label version with pre-extracted CLIP features Radford et al. (2021). Each image is associated with multiple object labels (e.g., `person`, `car`, `dog`). The dataset contains 80 labels and exhibits high label co-occurrence and imbalance. **Yeast** is a classic bioinformatics dataset introduced by Elisseeff & Weston (2001) for protein function prediction. It consists of 2,417 samples and 14 labels, with moderate label density. Each instance represents a gene with various expression-based features. **Emotions** is a music-related dataset that maps songs to a set of emotion labels such as `happy`, `sad`, and `relaxing`. It contains 593 instances and 6 labels. This dataset is relatively balanced and is commonly used in multi-label benchmarking Trohidis et al. (2008).

### 5.2 EVALUATION METRICS

We evaluate the performance using the following five metrics:

**Empirical Coverage (EC)** is the proportion of true labels captured by the prediction set, averaged across all validation instances:

$$\text{Empirical Coverage} = \frac{1}{N} \sum_{i=1}^{N} \frac{|\hat{Y}(x_i) \cap Y_i|}{\max(1, |Y_i|)}.$$

**Average Set Size:** The average number of labels predicted per instance:

$$\text{Set Size} = \frac{1}{N} \sum_{i=1}^{N} |\hat{Y}(x_i)|.$$

**Marginal Coverage (MC)** is the average probability that a true label is included in the predicted set, computed separately for each label and averaged:

$$\text{Marginal Coverage} = \frac{1}{L} \sum_{j=1}^{L} \mathbb{P}\left( j \in \hat{Y}(x) \,\Big|\, j \in Y \right).$$

and **Macro-F1 Score** averages F1 of all labels (i.e., binary predictions v. ground truth per label). Each of these metrics is applied to all individual classifiers and to aggregated predictions of ensembles (see §3.2).

## 5.3 TRAINING SETUP

Datasets are split into 60% training and 40% test sets; the test set is further divided equally into calibration and validation subsets (20% each). We adopt a **binary relevance** approach, training one binary classifier per label. For ensembles, $M = 3$ or $5$ base models are trained on bootstrap samples, each independently calibrated using conformal prediction (CP). Homogeneous ensembles use repeated instances of the same model; heterogeneous ensembles combine LR, SGD, MLP, and others. Aggregation is performed via majority voting, probability averaging, F1-weighted voting (based on validation F1 per label), or stacked ensembling with a LR meta-learner calibrated post hoc. Linear models (LR, SGD) are trained using `scikit-learn` with standard settings (log loss, Platt scaling for SGD). Neural models (MLP, RNN, Transformer, Mixer) are implemented in PyTorch. MLP uses one hidden layer (256 ReLU units); RNN is a unidirectional LSTM with hidden size 128; Transformer has two attention layers (4 heads); MLP-Mixer uses standard token/channel mixing and GELU activations. All deep models are trained with Adam (learning rate $10^{-3}$), binary cross-entropy loss, batch size 128, for 10 epochs. 30% of each bootstrap sample is used for calibration. All models consume CLIP embeddings as input. Training is conducted on a single NVIDIA GPU, and results are averaged over 3 random seeds for robustness.

## 5.4 EXPERIMENTAL RESULTS

Table 1 presents a comparative evaluation of our ECP methods across the three MLC datasets (Emotions, Yeast, COCO), compared against non-conformal baselines, single-model conformal predictors (CP), and post-hoc conformalized ensembles. Performance is assessed using empirical coverage (EC), marginal coverage (MC), average prediction set size, and macro-F1 (runtime analysis in A.5).

Table 1: Performance comparison across Emotions, Yeast, and COCO for multi-label classification. EC: Empirical Coverage, MC: Marginal Coverage. Best CP-based results per dataset in **bold**.

| Method | Emotions | | | Yeast | | | COCO | | |
|---|---|---|---|---|---|---|---|---|---|
| | EC/MC | Set Size | F1 | EC/MC | Set Size | F1 | EC/MC | Set Size | F1 |
| *Non-Conformal* | | | | | | | | | |
| BR (LR) | – | – | 0.615 | – | – | 0.350 | – | – | 0.698 |
| Ensemble (LR) | – | – | 0.349 | – | – | 0.349 | – | – | 0.696 |
| CLIP-RNN | – | – | – | – | – | – | – | – | 0.700 |
| Label Bagging (10/40) | – | – | – | – | – | – | – | – | 0.483 / 0.694 |
| *Single CP* | | | | | | | | | |
| CP (LR) | 0.891/0.877 | 3.45 | 0.647 | 0.8916/0.883 | 10.61 | 0.453 | 0.910/0.900 | 8.56 | 0.525 |
| CP (MLP) | 0.903/0.893 | 4.42 | 0.543 | 0.903/0.893 | 10.54 | 0.468 | 0.909/0.899 | 9.34 | 0.510 |
| CP (SGD) | 1.000/1.000 | 6.00 | 0.469 | 0.898/0.871 | 10.62 | 0.456 | 0.911/0.901 | 8.67 | 0.528 |
| CP (RNN) | – | – | – | – | – | – | 0.912/0.901 | 8.08 | 0.542 |
| *Post-hoc CP Ensemble* | | | | | | | | | |
| Het. Ensemble - CP | 0.966/0.894 | 3.49 | 0.629 | 0.904/0.893 | 10.71 | 0.463 | 0.905/0.892 | 7.83 | 0.548 |
| Multiple MLPs - CP | 0.890/0.878 | 4.49 | 0.515 | – | – | – | – | – | – |
| Stacked Het. - CP | 0.887/0.878 | 3.26 | 0.645 | 0.874/0.870 | 10.17 | 0.466 | – | – | – |
| *ECP (ours)* | | | | | | | | | |
| Hom. CP (MLP-WA) | 0.872/0.863 | **3.17** | 0.647 | 0.913/0.893 | 10.38 | **0.471** | 0.904/0.889 | 7.96 | 0.543 |
| Hom. CP (LR-MV) | – | – | – | 0.909/0.873 | 10.35 | 0.4635 | 0.896/0.897 | **7.16** | 0.567 |
| Het. CP (MV/WV) | – | – | – | 0.910/0.870 | **10.10** | **0.471** | 0.906/0.887 / 0.887/0.858 | 7.68 / 7.22 | 0.555 / **0.575** |
| Stacked Het. CP | 0.899/0.883 | 3.40 | **0.660** | – | – | – | – / – | – / – | – / – |

Our ECP methods consistently outperform single-model CP and post-hoc ensembles across most metrics. Table 1 groups methods into four categories: *Non-Conformal* (plain classifiers, no uncertainty quantification), *Single CP* (label-wise CP on individual models), *Post-hoc CP Ensemble* (ensemble first, then conformalize), and *ECP (ours)* (conformalize base models individually, then aggregate). Within each block, we report empirical coverage (EC), marginal coverage (MC), average set size, and macro-F1. Note that rows such as *Het. CP (MV/WV)* report results for two separate ensemble strategies: majority voting (MV) and F1-weighted voting (WV), with the left/right values corresponding to each variant.

**On the Emotions dataset**, our stacked heterogeneous ECP achieves the best macro-F1 (0.660), surpassing both single-model CP and post-hoc ensembles, while keeping the set size compact (3.40).

Homogeneous MLP ensembles with weighted averaging (WA) are also competitive (0.647 F1, smallest set size 3.17), showing the benefit of probabilistic aggregation. In contrast, CP (SGD) reaches perfect coverage (EC = 1) but at the cost of very large sets (average size 6.00), which in fact equals the total number of possible labels in the Emotions dataset, leading to poor F1 and illustrating the conservativeness of single-model CP.

**For the Yeast dataset**, our methods again provide improvements. Both the homogeneous MLP ensemble (ECP, WA) and the heterogeneous ensemble (ECP, MV) deliver the top macro-F1 (0.471), with the heterogeneous variant producing more compact sets (10.10 vs. 10.38). Our homogeneous MLP-WA model also achieves the highest EC (0.913) and MC (0.893), indicating strong reliability. Post-hoc CP ensembles improve upon single-model CP, but do not reach the efficiency or predictive strength of ECP.

**The COCO dataset**, with its high label cardinality and imbalance, is the most challenging. Here, our heterogeneous ECP with F1-weighted voting (WV) achieves the highest macro-F1 (0.575) while maintaining compact sets (7.22). Our homogeneous LR ensemble with majority voting (MV) yields the smallest prediction sets overall (7.16) with competitive F1 (0.567). Other ECP variants (Hom. WA, Het. MV) strike balanced trade-offs between accuracy and compactness. In all cases, our ECP methods maintain valid coverage, while post-hoc ensembles and single CP are less effective.

**Summary of takeaways.** (i) *ECP consistently improves predictive performance and efficiency over single-model CP and post-hoc ensembles.* (ii) *Heterogeneous ensembles generally outperform homogeneous ones* due to increased model diversity. (iii) *Aggregation strategies matter:* MV, WV, and WA each provide benefits, with stacked ensembles offering the highest gains when sufficient validation data is available. (iv) Across datasets, *our methods achieve either the best F1 or the most compact sets, while preserving coverage*, demonstrating that ECP is a robust framework for uncertainty-aware multilabel classification.

## 5.5 ABLATION STUDY

To further isolate the effects of individual design choices in our framework, such as ensemble size, model diversity, aggregation strategy, and coverage parameters, we report targeted ablation studies in the Appendix A. On the COCO dataset, we find that increasing the ensemble size up to 5 models improves macro-F1 and stabilizes empirical coverage, without increasing prediction set size. Heterogeneous ensembles outperform homogeneous ones possibly due to architectural diversity. Varying the miscoverage rate $\alpha$ confirms the expected accuracy–coverage trade-off, with higher $\alpha$ values leading to smaller, more precise prediction sets at the expense of coverage. We also repeated experiments across five random seeds, with results summarized in Table 2, confirming that our ensembles yield consistent improvements and remain robust to training variability. These findings offer practical guidance for tailoring ECP systems to specific performance requirements.

Table 2: Macro-F1 (mean ± std) across five runs.

| Dataset | Method | Macro-F1 |
|---|---|---|
| COCO | Single CP (MLP) | 0.542 ± 0.001 |
| | ECP (ours) | **0.558 ± 0.003** |
| Emotions | Single CP (MLP) | 0.554 ± 0.015 |
| | ECP (Stacked Ensemble) | **0.647 ± 0.024** |
| Yeast | Single CP (MLP) | 0.466 ± 0.007 |
| | ECP (Het. Ensemble) | **0.467 ± 0.004** |

## 6 CONCLUSION

In this work, we systematically studied ensemble-based conformal prediction (ECP) methods for multilabel classification, combining the predictive strength of ensembles with the formal uncertainty guarantees of conformal prediction. To the best of our knowledge, this is the first such study. By adapting and evaluating multiple aggregation strategies across three benchmarks, we showed

that ECP improves F1 over single-model and post-hoc conformal baselines while maintaining valid empirical and marginal coverage and producing smaller prediction sets. Our multi-seed evaluation further demonstrated that ensembles provide more stable and robust performance. These findings establish ECP as a practical and flexible framework for uncertainty-aware multilabel learning. Future work will investigate richer base architectures, adaptive calibration for rare labels, and efficient conformalization strategies to extend ECP to extreme-label and large-scale applications.

## REPRODUCIBILITY STATEMENT

Our results are designed to be fully reproducible. All datasets used in this work (MS-COCO, Yeast, Emotions) are publicly available and described in Section 5.1. Model architectures, training setups, and ensemble configurations are specified in Section 5.3 and Section 5. Evaluation metrics are formally defined in Section 5.2. To assess robustness, we report results averaged over multiple random seeds and provide standard deviations in Appendix A.6. Extended ablation studies (Appendix A.1) further document the sensitivity of our framework to ensemble size, miscoverage rate, and model diversity. An anonymous implementation, including training scripts and conformal calibration routines, is submitted in the supplementary materials to facilitate exact reproduction.

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

## A  APPENDIX

### A.1  ABLATION STUDIES

To isolate and analyze the contributions of individual components within our proposed ECP framework, we conduct comprehensive ablation experiments using the COCO dataset. Specifically, we investigate the impact of ensemble size, aggregation strategy (majority voting versus probability averaging), model diversity (homogeneous versus heterogeneous ensembles), and the sensitivity of coverage guarantees to the specified miscoverage rate ($\alpha$).

### A.2  IMPACT OF ENSEMBLE SIZE

We first examine the effect of varying the ensemble size $M$ from 1 (single-model baseline) to 10 using homogeneous ensembles of LR classifiers. Figure 1 illustrates the changes in empirical coverage, marginal coverage, macro-F1, and average prediction set size as the ensemble size increases.

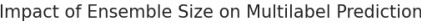

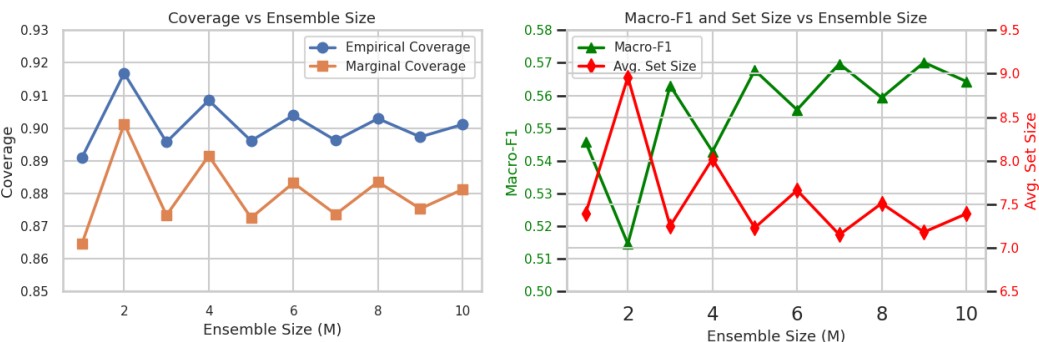

Figure 1: Impact of ensemble size on multilabel prediction performance on the COCO dataset. Left: Empirical and marginal coverage increase moderately with ensemble size, peaking around $M = 2$–4. Right: Macro-F1 improves steadily with larger ensembles while average prediction set size remains compact and stable, showing strong trade-offs between accuracy and efficiency.

Table 3 summarizes detailed numerical results. Increasing the ensemble size from 1 to 5 models leads to significant improvements in macro-F1 scores (approximately 2.2 percentage points, from 0.5458 to 0.5677) and maintains a relatively stable empirical coverage around 0.89–0.91. Additionally, the average prediction set size remains compact, decreasing slightly from 7.39 to 7.23 labels per instance. Beyond 5 models, performance gains become marginal, indicating a saturation effect at approximately $M = 5$.

Thus, for practical considerations balancing predictive accuracy, coverage, and computational cost, an ensemble size between 3 and 5 is recommended.

Table 3: Detailed results of varying ensemble size ($M$) on COCO dataset.

| $M$ | EC | Avg. Size | Macro-F1 | MC |
|---|---|---|---|---|
| 1 | 0.8909 | 7.39 | 0.5458 | 0.8645 |
| 2 | 0.9168 | 8.96 | 0.5145 | 0.9013 |
| 3 | 0.8957 | 7.25 | 0.5630 | 0.8733 |
| 4 | 0.9085 | 8.02 | 0.5428 | 0.8915 |
| 5 | 0.8961 | 7.23 | 0.5677 | 0.8725 |
| 6 | 0.9039 | 7.66 | 0.5556 | 0.8832 |
| 7 | 0.8962 | 7.15 | 0.5697 | 0.8736 |
| 8 | 0.9029 | 7.51 | 0.5593 | 0.8835 |
| 9 | 0.8973 | 7.18 | 0.5701 | 0.8753 |
| 10 | 0.9011 | 7.39 | 0.5643 | 0.8812 |

## A.3    SENSITIVITY TO MISCOVERAGE RATE

To evaluate the robustness of our ECP framework under varying coverage requirements, we analyze its sensitivity to the target miscoverage rate $\alpha$. We vary $\alpha$ across a range of values {0.01, 0.05, 0.10, 0.20}, which correspond to target coverage levels of 99%, 95%, 90%, and 80%, respectively. Table 4 presents the resulting empirical coverage and macro-F1 scores for each setting.

As expected, reducing the target coverage level (increasing $\alpha$) results in decreased empirical coverage. However, this trade-off allows for more selective predictions, which significantly improves the macro-F1 score. For example, increasing $\alpha$ from 0.01 to 0.20 improves macro-F1 from 0.28 to 0.67, illustrating a strong accuracy-coverage trade-off. These results emphasize the importance of tuning $\alpha$ based on the application's tolerance for errors versus the need for precision.

Table 4: Effect of target miscoverage rate ($\alpha$) on empirical coverage and macro-F1.

| $\alpha$ | Target Coverage | Empirical Coverage | Macro-F1 |
|---|---|---|---|
| 0.01 | 0.99 | 0.9877 | 0.2825 |
| 0.05 | 0.95 | 0.9399 | 0.4874 |
| 0.10 | 0.90 | 0.8878 | 0.5875 |
| 0.20 | 0.80 | 0.7887 | 0.6700 |

## A.4    HOMOGENEOUS VS. HETEROGENEOUS ENSEMBLES

To understand the impact of model diversity on conformal ensemble performance, we compare homogeneous ensembles (composed of identical model types, e.g., multiple MLPs) with heterogeneous ensembles (comprising diverse architectures such as LR, SGD, and MLP). This comparison is conducted across all three datasets under CP with majority voting or weighted aggregation.

Results in Table 1 suggest that heterogeneous ensembles generally offer improved performance in terms of macro-F1 while maintaining strong empirical and marginal coverage. For instance, on the Emotions dataset, the *Stacked Heterogeneous Ensemble* achieves the best overall F1 score (0.6596), outperforming both single-model CP and homogeneous CP ensembles. Similarly, on COCO, the heterogeneous ensemble (Het. (2) CP (WV)) obtains the highest F1 (0.5745) while maintaining a competitive set size (7.22) and adequate marginal coverage.

In contrast, homogeneous ensembles—such as MLP-based WA ensembles—tend to produce slightly larger prediction sets (e.g., Emotions: size 3.17, F1 0.6467), though they still yield competitive results when well-calibrated. These findings reinforce the hypothesis that diversity in base models helps mitigate correlated errors, leading to more compact and accurate prediction sets.

In summary, while homogeneous ensembles provide stable baselines, heterogeneous ensembles consistently achieve better accuracy–coverage trade-offs, especially when combined with voting or weighted aggregation mechanisms.

## A.5 Runtime and Computational Analysis

All experiments were conducted on a machine with an NVIDIA GeForce RTX 2080 Ti GPU (11GB VRAM), 64GB RAM, and Ubuntu 20.04. On the COCO dataset, single-model CP completes calibration and inference in approximately 16 minutes. Ensemble variants with 5 models require around 22 minutes, reflecting a 38% increase in runtime. This overhead scales roughly linearly with ensemble size but remains practical for configurations with 3–5 models. Given the gains in coverage and predictive robustness, the added cost is considered acceptable for most applications. Parallelization can further improve efficiency.

## A.6 Statistical Significance

To evaluate whether the performance improvements from ensemble-based conformal predictors are statistically reliable, we repeated experiments across five random seeds for all three datasets. We report the mean and standard deviation (std) of the macro-F1 score in Table 5. The standard deviation reflects the variability across runs: smaller std indicates more stable performance. In addition, for Emotions and Yeast we also report coverage and average set size (Table 6), since these metrics are particularly relevant for smaller-scale MLC benchmarks.

Table 5: Macro-F1 (mean ± std) across five runs.

| Dataset | Method | Macro-F1 |
|---|---|---|
| COCO | Single CP (MLP) | 0.542 ± 0.001 |
| | ECP (ours) | **0.558 ± 0.003** |
| Emotions | Single CP (MLP) | 0.554 ± 0.015 |
| | ECP (Stacked Ensemble) | **0.647 ± 0.024** |
| Yeast | Single CP (MLP) | 0.466 ± 0.007 |
| | ECP (Het. Ensemble) | **0.467 ± 0.004** |

Table 6: Extended reliability metrics (mean ± std) across five runs for Emotions and Yeast.

| Dataset | Method | Coverage | MC | Set Size | F1 |
|---|---|---|---|---|---|
| Emotions | Single CP (MLP) | 0.8641 ± 0.0255 | 0.8617 ± 0.0231 | 3.97 ± 0.23 | 0.5542 ± 0.0151 |
| | ECP (Stacked) | 0.8641 ± 0.0419 | 0.8678 ± 0.0345 | **3.24** ± 0.30 | **0.6467** ± 0.0238 |
| Yeast | Single CP (MLP) | 0.9055 ± 0.0063 | 0.9025 ± 0.0093 | 10.55 ± 0.14 | 0.4661 ± 0.0071 |
| | ECP (Het.) | 0.9103 ± 0.0082 | 0.8851 ± 0.0183 | **10.36** ± 0.25 | **0.4667** ± 0.0035 |

Overall, the results confirm that ensemble-based CP improves predictive performance in Emotions substantially (F1 gain of +0.09, with smaller sets), achieves moderate but consistent gains on COCO, and maintains competitive performance on Yeast while slightly improving prediction set compactness. On COCO, a Wilcoxon signed-rank test yielded a $p$-value of 0.062, suggesting marginal significance. For Emotions, the gain is well beyond the baseline's variability, confirming a robust improvement. On Yeast, the improvements are minor in F1 but demonstrate the stability of ensemble calibration.

## A.7 Limitations and Future Work

While our method improves both uncertainty quantification and predictive performance in multi-label classification, it also opens several avenues for future improvement. First, our current framework treats labels independently through the binary relevance assumption. This simplifies calibration but ignores structured label dependencies; as the label space grows, this independence can lead to inefficiencies and redundant prediction sets. Extending ensemble conformal prediction to incorporate correlations (e.g., via graphical models, label hierarchies) is an important direction. Second, ensemble methods add computational overhead, both in training multiple base models and in conformal calibration for each label. Although ensembles of moderate size are tractable, scaling to

large base models or extreme multi-label settings (hundreds or thousands of labels) may require more efficient strategies such as pruning, model distillation, or approximate calibration. Third, performance depends on the quality of base models. Miscalibration, especially for rare labels, can propagate through the conformal procedure. More adaptive calibration schemes, such as label-wise adjustments or focal loss–based training, could improve reliability. Future work will also explore extending our framework beyond binary relevance to structured multi-label outputs, investigating ensemble methods tailored for extreme label spaces, and developing computationally efficient conformalization techniques suitable for real-time or large-scale applications.

## A.8 PROOF OF LEMMA 1

Fix a label $\ell \in \{1, \ldots, L\}$. For each base model $m \in \{1, \ldots, M\}$, let

$$X_m^{(\ell)} := \mathbf{1}\{\ell \in \widehat{Y}^{(m)}(x)\}$$

denote the indicator that model $m$ includes label $\ell$ in its prediction set.

**Assumption.** By the label-wise marginal coverage of each conformal predictor,

$$p_m := \mathbb{P}(\ell \in \widehat{Y}^{(m)}(x) \mid \ell \in Y) = \mathbb{P}(X_m^{(\ell)} = 1 \mid \ell \in Y) \geq 1 - \alpha$$

for all $m \in \{1, \ldots, M\}$, and we assume that the indicators $X_1^{(\ell)}, \ldots, X_M^{(\ell)}$ are independent conditional on $\{\ell \in Y\}$.

**Coupling construction.** Let $U_1, \ldots, U_M$ be i.i.d. $\mathrm{Unif}(0,1)$ random variables independent of $(x, Y)$. Conditional on $\{\ell \in Y\}$ we may realize the Bernoulli indicators as

$$X_m^{(\ell)} = \mathbf{1}\{U_m \leq p_m\}, \qquad m = 1, \ldots, M,$$

which preserves both the Bernoulli marginals and conditional independence. Define auxiliary Bernoulli variables

$$B_m := \mathbf{1}\{U_m \leq 1 - \alpha\}, \qquad m = 1, \ldots, M.$$

Then, conditional on $\{\ell \in Y\}$, the variables $B_1, \ldots, B_M$ are i.i.d. $\mathrm{Bernoulli}(1 - \alpha)$.

**Stochastic dominance.** Since $p_m \geq 1 - \alpha$ for all $m$, we have

$$\{U_m \leq 1 - \alpha\} \subseteq \{U_m \leq p_m\} \quad \Rightarrow \quad B_m \leq X_m^{(\ell)} \quad \text{almost surely.}$$

**Summation and probability bound.** Let $S_X := \sum_{m=1}^{M} X_m^{(\ell)}$ and $S_B := \sum_{m=1}^{M} B_m$. The pointwise inequality implies $S_B \leq S_X$ almost surely, so for any voting threshold $k \in \{1, \ldots, M\}$,

$$\mathbb{P}(S_X \geq k \mid \ell \in Y) \geq \mathbb{P}(S_B \geq k \mid \ell \in Y) = \sum_{r=k}^{M} \binom{M}{r} (1 - \alpha)^r \alpha^{M-r},$$

where the equality follows because $S_B \mid \{\ell \in Y\} \sim \mathrm{Binomial}(M, 1 - \alpha)$.

For unanimity voting ($k = M$), this gives

$$\mathbb{P}\left( \sum_{m=1}^{M} X_m^{(\ell)} = M \,\Big|\, \ell \in Y \right) \geq (1 - \alpha)^M.$$

$\square$

## A.9 PROOF OF THEOREM 1 (UNANIMITY ENSEMBLE COVERAGE)

Consider an ensemble using unanimity voting: label $\ell$ is included in $\widehat{Y}^{\mathrm{ens}}(x)$ if and only if all $M$ base models include it, i.e.,

$$\ell \in \widehat{Y}^{\mathrm{ens}}(x) \quad \Longleftrightarrow \quad \sum_{m=1}^{M} X_m^{(\ell)} = M.$$

Conditioning on the event $\{\ell \in Y\}$ and applying Lemma 1 with $k = M$ yields

$$\mathbb{P}(\ell \in \widehat{Y}^{\mathrm{ens}}(x) \mid \ell \in Y) = \mathbb{P}\left( \sum_{m=1}^{M} X_m^{(\ell)} = M \,\Big|\, \ell \in Y \right) \geq (1-\alpha)^M.$$

The upper bound of 1 is trivial since probabilities cannot exceed 1. Therefore,

$$(1-\alpha)^M \leq \mathbb{P}(\ell \in \widehat{Y}^{\mathrm{ens}}(x) \mid \ell \in Y) \leq 1.$$

$\square$

EXTENSION TO GENERAL VOTING THRESHOLDS

For a general voting threshold $k \in \{1, \dots, M\}$ (i.e., include label $\ell$ if at least $k$ models vote for it), Lemma 1 directly gives

$$\mathbb{P}\big(\ell \in \widehat{Y}^{\mathrm{ens}}(x) \mid \ell \in Y\big) = \mathbb{P}\left( \sum_{m=1}^{M} X_m^{(\ell)} \geq k \,\Big|\, \ell \in Y \right) \geq \sum_{r=k}^{M} \binom{M}{r} (1-\alpha)^r \alpha^{M-r}.$$

In particular, for majority voting with $k = \lceil M/2 \rceil$, this binomial tail lower bound is typically much larger than $(1-\alpha)^M$ and, whenever $1 - \alpha > 1/2$, it tends to 1 as $M$ grows.

### A.10 PRACTICAL GUIDANCE FOR AGGREGATION RULES

In this section we provide a concise practitioner-oriented guide for choosing between majority voting (MV), probability averaging (PA), and F1-weighted aggregation (WV).

Our results show that the relative effectiveness of MV, PA, and WV depends on three factors: (i) calibration quality of the ensemble members, (ii) diversity and performance variability across models, and (iii) the size of the validation set available for estimating weights. The following recommendations provide a simple rule-of-thumb for choosing an aggregation method in practice.

**Probability averaging (PA)** is preferred when ensemble members produce reasonably calibrated probabilities (e.g., after temperature scaling or through logistic regression heads) and when their predictive performance is relatively similar. In these settings, as seen for the homogeneous MLP ensembles on EMOTIONS and YEAST, PA yields stable predictive sets with good calibrated coverage and competitive macro-F1.

**Majority voting (MV)** is robust when model outputs are poorly calibrated or when only hard predictions are available. This was particularly effective for heterogeneous ensembles on COCO, where the underlying models exhibit disparate calibration quality (e.g., SGD vs. MLP). MV smooths out miscalibration and tends to preserve empirical coverage even under substantial variability across members.

**F1-weighted aggregation (WV)** is most beneficial when ensemble members differ markedly in predictive strength and a sufficiently large validation set is available to reliably estimate per-model F1 weights. This scenario occurs in COCO, where model performance differences are large; WV leverages stronger models more heavily and improves macro-F1 while preserving coverage.

Table 7 provides a concise decision chart summarizing these recommendations.

Table 7: Practical guide for choosing an aggregation rule based on data size, model diversity, and calibration quality.

| Scenario | Data size | Model diversity | Calibration quality | Recommended rule |
|---|---|---|---|---|
| Models are similarly strong and calibration is good | Medium–large | Low–moderate | Good | Probability averaging (PA) |
| Scores are poorly calibrated or only hard labels available | Any | Any | Poor / unknown | Majority voting (MV) |
| Large gaps in model predictive performance | Medium–large | Any | OK–good | F1-weighted aggregation (WV) |
| Limited validation data for weight estimation | Small | Low–moderate | Unknown | Majority voting (MV) |

## A.11 COVERAGE ANALYSIS BY LABEL FREQUENCY

To complement aggregate metrics such as macro-F1 and exact match, we provide a more fine-grained analysis of marginal coverage. Figure 2 reports marginal coverage by label-frequency bucket (left) and cumulative per-label coverage (right), across all methods evaluated in our study.

Labels are grouped into frequency buckets based on their prevalence in the training data: $\leq 1\%$, 1–5%, 5–20%, and >20%. These buckets reflect the long-tailed nature of typical multilabel datasets. The cumulative plots sort labels by their marginal coverage, making deviations from the coverage target $1 - \alpha$ directly visible.

Overall, all methods satisfy the coverage target on average. Single-model CP (a,b) achieves slightly higher coverage for rare labels but does so at the cost of substantially larger prediction sets, as reported in the main results table. Post-hoc CP (c,d) performs similarly but with slightly reduced variance. In contrast, both ensemble-based methods (e–h) yield more uniform coverage across labels and frequency buckets, while producing significantly more compact prediction sets and improved macro-F1 scores. The heterogeneous ensemble (g,h) in particular achieves strong calibration even on infrequent labels ($\leq 1\%$), indicating improved robustness without sacrificing coverage.

These plots illustrate that ensemble methods, while slightly under-covering in some low-frequency cases, maintain coverage close to the target while substantially improving efficiency.

## A.12 EMPIRICAL ANALYSIS OF ENSEMBLE CORRELATION AND COVERAGE

To better understand the reliability of the ensemble-based conformal prediction system, we perform an empirical analysis of (i) the effective correlation among ensemble members and (ii) the relationship between the observed marginal coverage and the theoretical binomial lower bound.

**Effective ensemble correlation.**    For each label, we compute the pairwise correlation between the binary conformal inclusion decisions produced by all model pairs. Averaging these values yields an *effective per-label ensemble correlation*. Lower correlations indicate greater model diversity, which generally leads to more robust ensemble behavior; higher correlations indicate that models behave similarly and provide less independent information to the ensemble.

**Histogram of correlations.**    Figure 3 shows the empirical distribution of effective ensemble correlations across labels for both the heterogeneous and homogeneous ensembles. The heterogeneous ensemble displays substantially lower correlations, reflecting higher model diversity, while the homogeneous ensemble exhibits strong clustering near correlations of 0.7–0.9, consistent with the use of models of identical architecture and training procedure. These distributions help characterize the effective independence of the ensemble members and provide a diagnostic for understanding the resulting coverage behavior.

## A.13 USE OF LARGE LANGUAGE MODELS (LLMS)

During manuscript preparation, ChatGPT (GPT-5) was used to aid in phrasing and grammar polishing. All research ideas, methodological developments, experiments, and analysis were conceived, implemented, and validated entirely by the authors. The authors take full responsibility for the content of the paper.

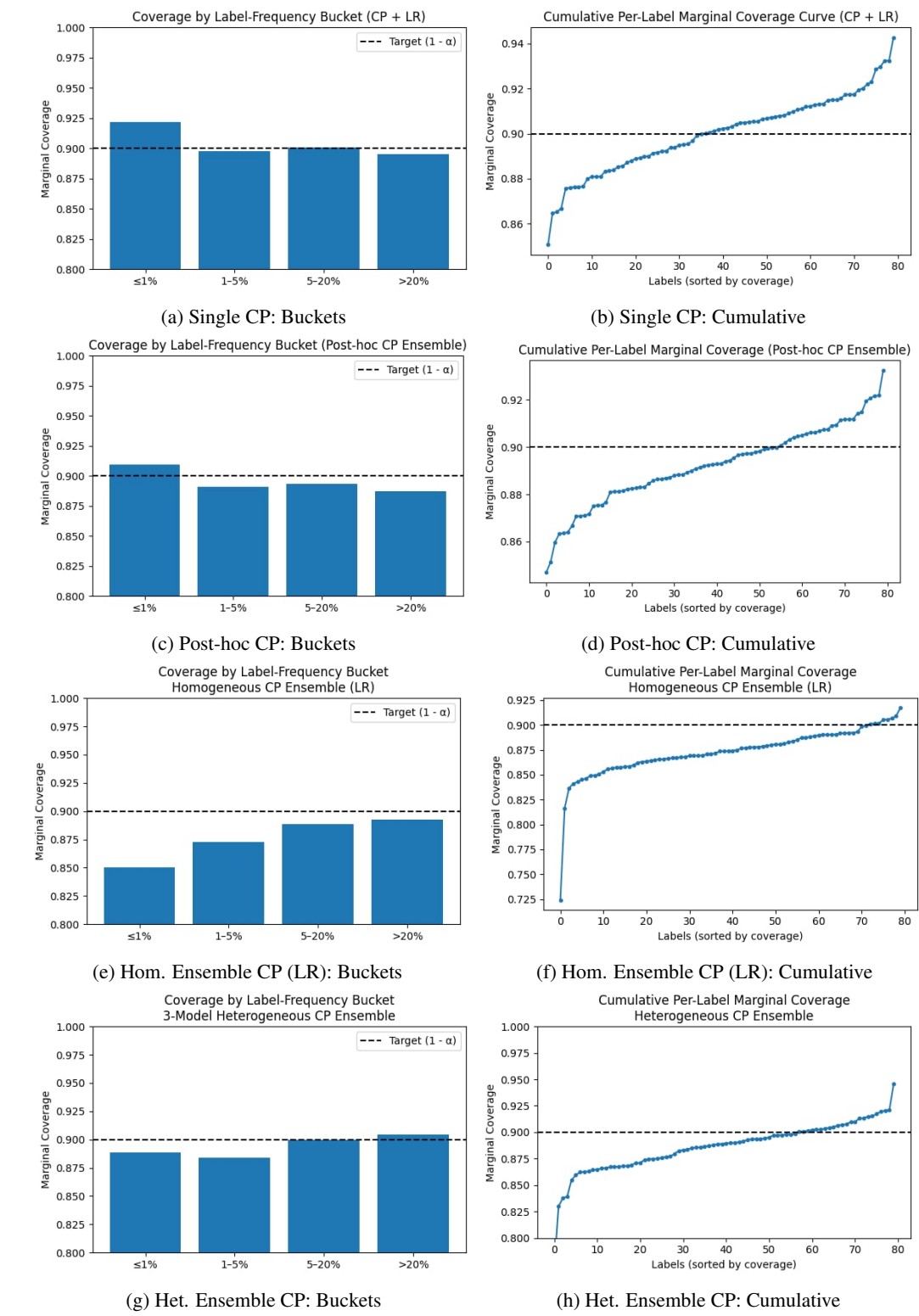

(a) Single CP: Buckets

(b) Single CP: Cumulative

(c) Post-hoc CP: Buckets

(d) Post-hoc CP: Cumulative

(e) Hom. Ensemble CP (LR): Buckets

(f) Hom. Ensemble CP (LR): Cumulative

(g) Het. Ensemble CP: Buckets

(h) Het. Ensemble CP: Cumulative

Figure 2: Marginal coverage analysis for all methods. Although the plots are shown on a restricted range (0.80–1.00), which visually amplifies small differences, all methods achieve marginal coverage very close to the target level $1 - \alpha$. The single-model CP baseline (CP + LR) appears slightly higher due to this scaling, but the absolute differences across methods are small (typically within 1–2%). Importantly, our ensemble-based approaches achieve comparable marginal coverage while producing substantially tighter prediction sets and higher macro-F1 scores (see Table 1), demonstrating improved efficiency without meaningful loss in coverage.

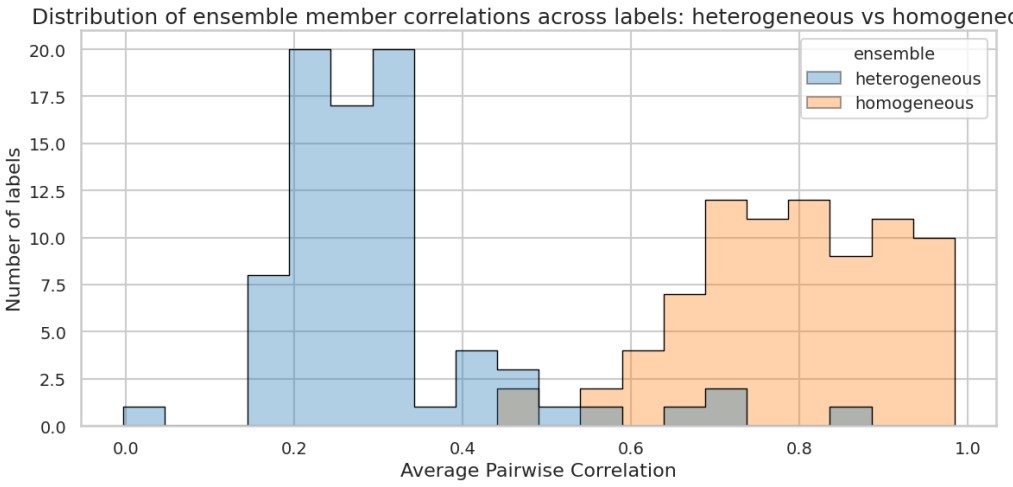

Figure 3: Empirical distribution of effective ensemble member correlations across labels for the heterogeneous (blue) and homogeneous (orange) ensembles. The y-axis reports the number of labels whose average model correlation falls within each bin. Lower correlations indicate more diverse model behavior, while higher correlations reflect more redundant ensemble predictions.

