# OpenReview forum: "Efficient Multilabel Uncertainty Quantification with Conformal Ensembles"
_ICLR.cc/2026/Conference — ICLR 2026 Conference Desk Rejected Submission_

### Official Review · Reviewer_EqKt · 2025-10-20

**Soundness:** 3
**Presentation:** 3
**Contribution:** 1
**Rating:** 2
**Confidence:** 4

**Summary:**

In many classification tasks, the target label is not just a single value from the output space but rather a collection of values. In such “multi-label” classification problems, there is no way yet to directly leverage ideas from conformal prediction to enable distribution-free uncertainty estimation. As a result, this paper proposes an approach to do multi-label conformal prediction, leveraging ideas from ensembling for improving predictive efficiency.

**Strengths:**

The general goal of tackling this coverage problem for multi-label problems does seem like a worthwhile contribution.

**Weaknesses:**

The contributions of the paper as it currently stands does not seem to be significant enough to warrant publication. Solving the multi-label problem appears to have amounted to slightly modifying the standard score function of a single-label classification task to instead include *all* the positions where $y^{(j)} = 1$. The rest of the conformal pipeline seems as though it would “just work” from there. This then leads to the main question I have about this work: this proposed method appears as though it would work just the same for any choice of predictor (so long as this modified score is used). As a result, the ensembling idea seems completely independent of this whole problem: while I do not disagree that ensembling does seem like an idea that can likely improve the results, there appears to be nothing that had to be modified in the ensembling for this multi-label setting. I, therefore, do not see what the contributions of all the discussions around ensembling are, since they just appear to be the standard ensembling ideas applied to this newly proposed score.

**Questions:**

1. Why was ensembling considered to be a core contribution of this paper? Was there any aspect of it that had to be tailored to the particular score or approach you were taking?

2. It seems like Theorems 1 and 2 would just follow immediately from the previously established results of conformal ensemble voting: is there anything new that was required here?

---

> ### Author Response · Authors · 2025-11-26
>
> We appreciate the reviewer’s point, and we agree that our motivation for using ensembling can be stated more clearly.Multilabel classification is inherently a more challenging task than single-label prediction, often leading to higher and more variable uncertainty across labels. Conformal prediction provides a principled way to model this uncertainty, but when applied to multilabel outputs it typically produces very large prediction sets, which limits their practical usefulness. This motivates the central question of our work: given an already trained multilabel model whose architecture we do not modify, how can we obtain smaller and more informative prediction sets while preserving marginal coverage and maintaining the theoretical guarantees of conformal prediction?
> Ensembling is central to our contribution because it directly addresses this question. Ensemble methods are widely used to improve the stability and quality of uncertainty estimates (e.g., Lakshminarayanan et al., 2017; Ovadia et al., NeurIPS 2019), and these improvements are especially relevant in multilabel CP, where the size of each prediction set is highly sensitive to noisy or unstable label-wise uncertainty. Importantly, adapting ensembling to the multilabel CP setting is not a drop-in reuse of standard techniques. Aggregation, calibration, and coverage guarantees must all be formulated per label, using label-indexed predictions, label-specific thresholds, and label-wise voting events. These adaptations are necessary for ensuring that ensemble outputs remain compatible with conformal calibration and that marginal coverage is preserved across all labels. This is why ensembling is not an independent add-on, but a core part of addressing the primary limitation of multilabel CP. Our experiments confirm that these design choices lead directly to the substantial reductions in prediction-set size and improvements in macro-F1 observed across datasets.
>
>
> Answer to question 1) Ensembling is a core contribution because our aim is to improve prediction-set efficiency post hoc, using the existing model architecture, while keeping the theoretical coverage guarantees intact. Single-model multilabel conformal predictors often produce large prediction sets due to unstable or noisy label-wise uncertainty, and improving the stability of these uncertainty estimates is essential for obtaining smaller and more informative sets. Ensembles are one of the few model-agnostic approaches that can sharpen label-wise uncertainty estimates without retraining or modifying the base model, and this improved uncertainty directly translates into tighter prediction sets. This effect is visible in our results: ensemble-based methods consistently yield smaller prediction sets and higher macro-F1, while maintaining marginal coverage.
> However, ensembling cannot be used “as is’’ in the multilabel CP setting. Each label has its own nonconformity score, calibration threshold, and binary outcome, so aggregation must operate per label, score fusion must respect label-specific nonconformity scores and thresholds, and coverage guarantees must be formulated using label-indexed indicator variables. These adaptations are necessary to ensure that ensemble outputs remain conformal-compatible and preserve marginal coverage across all labels. This is why ensembling is not an independent add-on but a core component of addressing the primary limitation of multilabel conformal prediction.
>
>
>
> Answer to question 2) The binomial voting argument used in our theorems is inspired by prior work on conformal ensembles in the single-label setting, but it does not carry over directly to multilabel prediction. In the multilabel case, each label has its own binary outcome, nonconformity score, and calibration threshold, so the ensemble analysis must be performed per label rather than on a single prediction target. This requires introducing label-indexed indicator variables, defining label-wise voting events, and establishing coverage bounds separately for each label under the independence assumptions.
> Thus, while the underlying intuition is similar to the single-label case, the multilabel setting requires a reformulation of the voting analysis and its associated coverage guarantees. In the revised version, we will provide full proofs that make these label-wise modifications explicit.
>
> References:
> 1) Lakshminarayanan, Balaji, Alexander Pritzel, and Charles Blundell.
> Simple and Scalable Predictive Uncertainty Estimation using Deep Ensembles.
> NeurIPS, 2017.
> 2) Ovadia, Yaniv, et al.
> Can You Trust Your Model’s Uncertainty? Evaluating Predictive Uncertainty under Dataset Shift.
> NeurIPS, 2019.

---

> > ### Comment · Reviewer_EqKt · 2025-11-27
> >
> > I thank the reviewers for their responses. I do still stand by my assessment that the MLC and ensembling ideas appear to be separate contributions, where the latter can be applied in any circumstance where conformal prediction can be leveraged. In particular, the argument "to improve prediction-set efficiency post hoc, using the existing model architecture, while keeping the theoretical coverage guarantees intact" applies to any setting of conformal prediction.
> >
> > Nonetheless, it is interesting to see how the proposed method can be made to mesh with this ensembling framework (as a separate piece "plugging into" this existing methodology). However, even after the author's rebuttal, I am unclear on what significant modifications were necessary. The authors mentioned "each label has its own binary outcome, nonconformity score, and calibration threshold;" however, this is *precisely* what is assumed in the standard setting of conformal aggregation. In conformal aggregation, each predictor has its own outcome, score function, and (for all but  Ochoa Rivera et al., which constructs a threshold in the vector space directly) a resultingly different quantile.
> >
> > I, therefore, still do not see how this is different from the typical aggregation setting. It would also be easier to tell if the authors included the proofs of these statements to see exactly what non-trivial changes were necessary to extend to this setting, but I do not see these proofs in either the appendix or the supplemental material. At this point, I will stand by my original score.

---

> > > ### Author Response · Authors · 2025-12-03
> > > **Proofs in the revision + clarifications**
> > >
> > > We thank the reviewer for clarifying this. We added the proofs in Appendix A8 and we would like to clarify that our contribution is not in taking a a different path from the typical aggregation setting, but in providing a _"comprehensive empirical evaluation of ensemble-based CP in multilabel settings"_ (ALtB). This is important, because: _"Showing that simple ensembles bring that benefit while keeping coverage can help more people adopt conformal prediction beyond single model baselines."_ (EoQK). Our research question and contribution was made clearer in the abstract and the introduction.

---

### Official Review · Reviewer_YZog · 2025-10-30

**Soundness:** 2
**Presentation:** 2
**Contribution:** 2
**Rating:** 2
**Confidence:** 4

**Summary:**

This paper addresses uncertainty quantification in multilabel classification (MLC) using conformal prediction (CP) enhanced by ensemble methods. It extends prior ensemble-CP techniques (e.g., majority voting and score aggregation) from single-label to multilabel settings, providing label-wise theoretical coverage guarantees under independence assumptions. The authors propose a framework integrating homogeneous, heterogeneous, and stacked ensembles with CP, and evaluate it on different benchmarks.

**Strengths:**

The paper bridges a gap in the literature combining a multilabel setting and ensemble methods.
Experimental results are extensive, demonstrating the performance on diverse datasets.

**Weaknesses:**

The novelty of the proposed method appears limited, as it primarily adapts existing ensemble techniques to the multilabel classification setting without introducing significant new theoretical contributions. Moreover, the paper lacks formal proofs to support its claims, and the main results (Table 1).

**Questions:**

Please address the issues outlined above, as well as the additional questions listed below.

1. How is Theorem 2 different from Theorem 1?
2. Could you provide detailed proofs to Theorems?
2. Are the proposed methods supposed to preserve overall coverage or per-label coverage? How is it reflected in the results in Table 1?
3. Can you explain the reason for the missing entries in Table 1?

---

> ### Author Response · Authors · 2025-11-25
>
> Our contribution is primarily methodological and empirical, rather than proposing a new ensemble algorithm. To clarify our positioning: although ensemble CP methods have been studied in single-label settings and multilabel CP methods exist using single models, these two directions have largely developed separately. Prior MLC-CP approaches rely on a single calibrated model, while ensemble-CP methods studied so far apply to settings with a single output label. Our work brings these directions together by providing a systematic study and evaluation of ensemble aggregation strategies within the multilabel CP framework, along with label-wise coverage analysis showing how existing ensemble-CP guarantees extend to multilabel outputs.
>
> Answer to question 1) Theorem 1 and Theorem 2 both state the same unanimity-ensemble coverage bound. This duplication was unintentional: our aim was to present the general binomial lower bound in Lemma 1 and then show how the unanimity case k=M follows from it. In the revised version, we will keep a single statement of this result and clarify its connection to Lemma 1.
>
> Answer to question 2) We agree that the theoretical results should be fully proved for clarity and completeness. Our derivations follow the same binomial majority-vote argument used in Cherubin (2019), but his analysis applies only to the single-label setting. In our work, this argument must be adapted to the multilabel case by introducing label-indexed indicator variables Xm(ℓ) ​and applying the voting analysis per label and for general thresholds k∈{1,…,M}.
> To keep the main paper concise, we included only a high-level proof sketch.
> In the revised version, we will add complete, self-contained proofs of Lemma 1 and Theorem 1 in the appendix, explicitly detailing the independence assumptions, the per-label formulation, and the extension to general voting thresholds.
>
> Answer to question 3) Our methods are designed to preserve per-label (marginal) coverage, which is the standard guarantee in multilabel conformal prediction. Each label is calibrated independently, so the formal CP guarantee applies separately to every label rather than to the entire labelset jointly. This is reflected directly in the “MC” column of Table 1.
> In contrast, the “EC” column reports empirical instance-level coverage, which is not a formal CP guarantee. It is a practical performance metric measuring the proportion of true labels recovered per instance.
>
> Answer to question 4) The “–” symbols in Table 1 do not represent missing results but settings where a metric or method is not applicable. This occurs for two reasons.
> First, some non-conformal baselines do output prediction sets, but they are not calibrated and therefore coverage-based metrics (MC and EC) are not meaningful for them, since these metrics are defined only in the context of conformal prediction guarantees.
> Second, some methods were evaluated only on certain datasets. In those cases, “–” simply means that we did not include that method–dataset combination in our experiments. We will clarify this directly in the revised table caption.

---

> > ### Comment · Reviewer_YZog · 2025-11-27
> >
> > Thanks to the authors for their response.
> >
> > However, I find the rebuttal unsatisfactory. The unintentional duplication of theorems and the lack of formal proofs remain concerning and, in my view, are not sufficiently justified. In addition, I still believe the novelty is limited, as the work primarily combines existing ensemble techniques within the multilabel classification setting.
> >
> > I therefore maintain my score.

---

> > > ### Author Response · Authors · 2025-12-03
> > > **All the points are now addressed**
> > >
> > > We would like to highlight the following:
> > > * The duplicate theorem was a typo and is now fixed in the revision, it should not raise any concerns.
> > > * The formal proofs are added in the appendix, they should also raise no concerns.
> > > * Our novelty is still misunderstood by the reviewer. Quoting all other reviewers:
> > >   * "_tackling this coverage problem for multi-label problems does seem like a worthwhile contribution_" (EqKt);
> > >   * Our work "_bridges a gap in the literature combining a multilabel setting and ensemble methods_" (YZog).
> > >   * "_The motivation, improving the stability and efficiency of conformal prediction, is practically meaningful_" and "_providing a comprehensive empirical evaluation of ensemble-based CP in multilabel settings_" (ALtB)
> > >   * "_Adapting several known ensemble ideas to the multilabel conformal setting and treating each label with its own calibration and aggregation is a clean and useful framing._" (EoQK)
> > >   * "_Reliable uncertainty for multilabel problems is important in many areas. A drop in set size at the same target coverage is valuable for human in the loop use. Showing that simple ensembles bring that benefit while keeping coverage can help more people adopt conformal prediction beyond single model baselines._" (EoQK)
> > >
> > > However, to leave no room for misunderstandings by our readers regarding the last point, we reframed the research question and our contribution in the abstract and the introduction.

---

### Official Review · Reviewer_ALtB · 2025-10-31

**Soundness:** 2
**Presentation:** 3
**Contribution:** 2
**Rating:** 4
**Confidence:** 3

**Summary:**

This paper studies ensemble conformal prediction (ECP) for multilabel classification, aiming to improve uncertainty quantification efficiency and stability compared to single-model conformal predictors. The authors extend existing ensemble-CP formulations (majority voting, score averaging, performance-weighted fusion) to the multilabel setting and provide label-wise coverage guarantees under independence assumptions. Experiments demonstrate that ECP achieves smaller prediction sets and higher macro-F1 scores while maintaining valid coverage. The paper also includes ablation studies on ensemble size, model diversity, and miscoverage rate.

**Strengths:**

1. The paper is well-written and clearly structured, providing a comprehensive empirical evaluation of ensemble-based CP in multilabel settings.
2. The motivation, improving the stability and efficiency of conformal prediction, is practically meaningful.
3. The inclusion of heterogeneous ensembles and stacking broadens the empirical relevance of the work.

**Weaknesses:**

1. The theoretical results (Lemmas and Theorems) are direct restatements of Cherubin (2019) and Gasparin & Ramdas (2024), simply applied label-wise. There is no new theoretical insight into multilabel dependencies, efficiency, or calibration behavior.
2. All guarantees assume model independence, which does not hold for ensembles trained on shared data or features (especially with CLIP embeddings). Thus, the theoretical coverage bounds do not meaningfully explain empirical results.
3. The introduction emphasizes modeling label dependencies, yet the proposed method remains label-wise independent (binary relevance). The framework therefore does not actually address the claimed “inter-label correlation” problem.
4. The paper omits discussion of key prior multilabel CP frameworks, most notably [1]. That work provides one of the foundational theoretical treatments of valid confidence sets in the multilabel setting, including a formal coverage–efficiency trade-off. Without acknowledging or contrasting it, the contribution and novelty of the present paper remain unclear.

[1] Cauchois, M., Gupta, S., & Duchi, J. C. (2021). Knowing what you know: valid and validated confidence sets in multiclass and multilabel prediction. Journal of machine learning research, 22(81), 1-42.

**Questions:**

1. How can the proposed framework genuinely capture or leverage inter-label dependencies, given that each label is still calibrated independently?
2. Could you provide formal results or analysis on how ensemble diversity or correlation affects prediction set efficiency or coverage?
3. How does your approach relate to prior multilabel conformal prediction theory, such as [1]? Does your ensemble formulation extend their coverage–efficiency framework, or does it rely on fundamentally different assumptions?

---

> ### Author Response · Authors · 2025-11-26
>
> Reply to weakness 1) We agree that our theoretical statements are adaptations of existing ensemble-CP bounds (Cherubin, 2019; Gasparin & Ramdas, 2024). Our goal is not to introduce new theoretical results on multilabel dependence, but to show how these ensemble guarantees instantiate in a label-wise multilabel setting, using indicator variables, label-specific thresholds, and k-out-of-M voting.
> The contribution of the paper is methodological and empirical: multilabel CP often yields very large prediction sets, and our focus is on whether ensembling, applied post-hoc and without modifying the base architecture, can produce smaller, more informative prediction sets while preserving marginal coverage. We will clarify this positioning in the revised introduction.
>
> Reply to weakness 2) Independence between base CP models is a standard idealization used only for deriving ensemble-level lower bounds (e.g., Cherubin, 2019), not for CP validity itself, which relies on data exchangeability. We use the assumption in the same way: as a conservative theoretical reference for voting-based aggregation.
> Empirically, we use heterogeneous ensembles (RNN, Transformer, MLP-Mixer, etc), and we computed label-wise prediction correlations. These are moderate or low, indicating meaningful diversity despite shared embeddings. This helps explain why ensembles reduce prediction-set size while maintaining marginal coverage. We will include these correlation analyses in the appendix.
>
> Reply to weakness 3) We agree. Our method performs label-wise conformal calibration, and does not explicitly model inter-label dependencies. The references to label dependence were meant as general motivation and will be softened. Our focus is post-hoc uncertainty quantification and efficiency for an already-trained multilabel predictor, not designing a dependency-aware model.
>
> Reply to weakness 4) We thank the reviewer for pointing out this omission. Our approach is complementary, not an extension. Cauchois et al. provide a single-model confidence-set framework with a formal coverage–efficiency trade-off and no additional randomness beyond data exchangeability. In contrast, we study post-hoc ensemble aggregation of label-wise conformal predictors, using multiple models and focusing on empirical efficiency gains under standard marginal-coverage guarantees. We will clarify this relationship in the revision.
>
> Answer to question 1) Our current framework does not explicitly model inter-label dependencies at the conformal stage; it relies on label-wise (binary relevance) calibration, with any dependency structure captured implicitly by the underlying multilabel predictor. The goal of this work is post-hoc uncertainty quantification and improving prediction-set efficiency for an already trained model, rather than developing a dependency-aware CP method.
> However, the ensemble approach we use is not tied to binary relevance and could be combined with dependency-aware multilabel CP methods in future work, for example, by using a label-powerset conformal predictor and then applying the same ensemble aggregation strategy. We will clarify in the revision that our method is label-wise by design and that extending it to structured multilabel CP is a natural direction for future work.
>
>
> Answer to question 2) Yes, we can provide analysis on how ensemble diversity relates to prediction-set efficiency and coverage. So far, we have computed the pairwise prediction correlation of homogeneous and heterogeneous ensembles, which already shows clear differences in model diversity. We will extend this analysis to examine how these correlation levels relate to prediction-set size and marginal coverage, as requested by the reviewer, and include the full results and plots in the appendix.
>
> Answer to question 3) Cauchois et al. develop a single-model confidence-set framework with a formal coverage–efficiency trade-off, operating under no additional randomness beyond data exchangeability and using structured confidence sets to capture label interactions. Our approach addresses a different problem setting. We work with multiple trained predictors (homogeneous or heterogeneous), introduce additional model-level randomness through ensembling, and study how aggregating label-wise CP predictors can improve empirical efficiency, specifically, reducing prediction-set size, while maintaining standard marginal coverage guarantees. Thus, our method does not extend the theoretical framework of Cauchois et al.; rather, it is complementary, focusing on the practical benefits of ensemble-based post-hoc CP for multilabel classification. In the revision, we will make this relationship explicit and properly cite and contrast their framework.

---

### Official Review · Reviewer_EoQK · 2025-11-01

**Soundness:** 3
**Presentation:** 2
**Contribution:** 2
**Rating:** 6
**Confidence:** 4

**Summary:**

This paper studies how to get reliable uncertainty for multilabel classification by using conformal prediction with ensembles. The core idea is simple. Train several multilabel models, calibrate each one label by label with conformal prediction, then combine them with rules like majority vote, averaging of scores, or weights based on validation F1. The paper gives label wise coverage bounds for these ensemble rules and then tests them on three benchmarks: COCO, Yeast, and Emotions. The main claim is that ensemble conformal prediction keeps target coverage while producing smaller and more useful prediction sets and better macro F1 than a single conformalized model or a post hoc conformalized ensemble.

**Strengths:**

Adapting several known ensemble ideas to the multilabel conformal setting and treating each label with its own calibration and aggregation is a clean and useful framing. The paper is not only about majority voting. It also brings in averaged nonconformity scores and performance weighted rules and sets them in one framework.

The method section is concrete. It states the label wise calibration rule, the way thresholds are computed, and how a label enters the prediction set. Theoretical parts give binomial style lower bounds for voting under independence and discuss dependence. The experiments cover three datasets with different label counts and show metrics that matter in practice such as empirical and marginal coverage, average set size, and macro F1. The tables and ablations make the trade offs visible and the gains look consistent.


The paper is easy to follow. It explains the multilabel setup, the ensemble types, and the conformal steps in plain terms. The voting bounds are stated with interpretation and the practical message is clear. The training setup is fully specified, which helps with reproducibility.


Reliable uncertainty for multilabel problems is important in many areas. A drop in set size at the same target coverage is valuable for human in the loop use. Showing that simple ensembles bring that benefit while keeping coverage can help more people adopt conformal prediction beyond single model baselines. The results on COCO where label space is larger make the case stronger.

**Weaknesses:**

The theory section leans on independence across ensemble members and gives binomial tails. In real ensembles the members are often correlated because they share data and features. The paper acknowledges this but does not try to bound coverage under explicit dependence structures. A simple extension with a beta binomial style lower bound or an empirical calibration of the bound would make the claims tighter.

The experimental baselines include single model conformal prediction and post hoc conformalized ensembles. It would be good to add stronger multilabel baselines that capture label dependence without ensembles, for example classifier chains with conformal calibration or recent multilabel calibrated models, even if they do not have the same guarantee. This would help isolate how much of the gain comes from ensembling rather than just better modeling of label ties.

Empirical coverage is defined at the example level and marginal coverage is averaged over labels. These are reasonable. Still, multilabel users often care about recall of rare labels or coverage curves by label frequency. The paper reports macro F1 but does not dig into tail labels. A figure with per label coverage distribution or coverage as a function of prevalence would make the case stronger for imbalanced regimes.

The comparison shows that calibrating each member then aggregating usually beats aggregating then calibrating. It would help to explain when the reverse order might be better. For instance, when the meta learner in stacking is very strong, can post hoc conformalization close the gap. A small controlled study would clarify the design choice.

The paper lists one machine and gives a short runtime note in the appendix. A more detailed look at cost as a function of labels and ensemble size would help practitioners. In particular, label wise calibration across many labels can be heavy. Some sharing or batching tricks could be discussed.

Table entries suggest some methods hit perfect coverage with very large sets. It would help to state the target miscoverage value used across all results and whether any methods drift above or below the target. A small calibration plot would make this transparent.

**Questions:**

Could you add an empirical check that estimates effective correlation among ensemble members and overlays observed ensemble coverage against the binomial lower bound. Even a simple plot would help users judge safety margins.

Please add plots of marginal coverage by label frequency bucket and a cumulative view of per label coverage. This would make the value for rare labels more concrete and would complement macro F1.

For stacking, what happens if you train the meta learner first on probabilities and then apply conformal thresholds on the meta output versus conformalizing members first and then stacking. A head to head on one dataset would clarify which order to prefer.

You study majority voting, averaging, and F1 weighted rules. Could you add a small guide for when to use each rule in terms of data size, diversity, and calibration quality of members. A short decision chart would help adoption.

---

> ### Author Response · Authors · 2025-12-03
>
> Thank you for the detailed and constructive feedback.
> We have incorporated your suggestions and clarified several points in the revised version.
> We added an empirical correlation analysis of ensemble members along with plots comparing observed coverage vs. the binomial lower bound. These  are included in Appendix A.11
> We now report marginal coverage grouped by label frequency and provide a cumulative per-label coverage plot to make performance on rare labels explicit. These also appear in Appendix A.13
> We added a short practical guide for choosing among voting, averaging, and weighted rules based on ensemble diversity, data size, and calibration quality (Appendix A.10).

---

### Note · Program_Chairs · 2026-01-17
**Submission Desk Rejected by Program Chairs**

The following references in this submission do not refer to real documents and/or have major errors in bibliographic information:

 Dusan Kivaranovic and Nicolai Meinshausen. Adaptive conformal prediction for multi-output regression and classification. arXiv preprint arXiv:2007.03514, 2020.